# The arabidopsis WAVE/SCAR protein BRICK1 associates with cell edges and plasmodesmata

**Zhihai Chi**[1,2], **Chris Ambrose**[1]*

**1** Department of Biology, The University of Saskatchewan, College of Arts and Science, Saskatoon, Canada, **2** State Key Laboratory of Plant Cell and Chromosome Engineering, Institute of Genetics and Developmental Biology, Chinese Academy of Sciences, Beijing, China

* jca790@usask.ca

## Abstract

Plasmodesmata are specialized structures in plant cell walls that mediate intercellular communication by regulating the trafficking of molecules between adjacent cells. The actin cytoskeleton plays a pivotal role in controlling plasmodesmatal permeability, but the molecular mechanisms underlying this regulation remain unclear. Here, we report that BRK1, a component of the WAVE/SCAR complex involved in Arp2/3-mediated actin nucleation, localizes to PD and primary pit fields in *A. thaliana* cotyledons, leaves, and hypocotyls. Using a BRK1-YFP reporter line, we detected BRK1 enrichment at cell edges and in primary pit fields, identified by regions of reduced propidium iodide staining. We also observed colocalization between BRK1-YFP and the plasmodesmatal callose stain aniline blue, further supporting BRK1's association with Plasmodesmata. Together, these findings suggest that the WAVE/SCAR complex participates in plasmodesmatal regulation by promoting ARP2/3-dependent actin filament branching at plasmodesmata, complementing the role of linear actin stabilization by formins.

## Introduction

Plasmodesmata are microscopic channels that traverse plant cell walls, establishing direct cytoplasmic connections between neighboring cells. These structures enable the symplastic movement of molecules between cells, allowing for the distribution of resources and signaling molecules essential for development and environmental response [1,2]. Plasmodesmata are lined by the plasma membrane and are typically traversed by an extension of the endoplasmic reticulum called a desmotubule. The diameter and permeability of plasmodesmata are highly dynamic and regulated to control what can pass through, ranging from ions and small molecules to larger macromolecules such as RNA and proteins. Plasmodesmatal permeability is measured as the size exclusion limit, referring to the physical size at which molecules are too large to pass through plasmodesmata. Plasmodesmatal permeability is regulated

**Data availability statement:** All original image files are available from the Zenodo database (doi.org/10.5281/zenodo.15839799).

**Funding:** NSERC Discovery Grant 2015–05938 The funders had no role in study design, data collection and analysis, decision to publish, or preparation of the manuscript.

**Competing interests:** The authors have declared that no competing interests exist.

primarily by deposition of callose (β-1,3-glucan) at the neck region surrounding the opening, where higher callose deposition reduces the size exclusion limit or occludes plasmodesmata.

Plasmodesmata are classified by their mechanism of formation. Primary plasmodesmata are formed during cytokinesis as ER tubules get trapped in the coalescing cell plate materials, while secondary plasmodesmata are formed after cytokinesis within existing cell walls [3,4]. Plasmodesmata are often clustered together in thin regions of the primary cell wall, termed pit fields [5,6]. In cells with secondary cell walls, deposition of the secondary wall is excluded from the primary pit field, forming a pit, where the former primary pit field is modified to form the pit membrane [7].

The actin cytoskeleton is a critical regulator of plasmodesmatal function [8]. Actin microfilaments (F-actin) have been shown to accumulate at or traverse plasmodesmata [9], and a number of actin binding proteins have been identified at plasmodesmata including nucleation components for both linear and branch actin (formins and the ARP2/3 complex; [10–13]), myosins [14–17], Actin-Depolymerizing Factor3 [18], the plant-specific NET family proteins [19], and a range of others via proteomic analysis [20].

These diverse roles of plasmodesma-associated actin-binding proteins suggests a sophisticated level of cytoskeletal coordination in regulating intercellular communication and maintaining plasmodesmatal functionality. However, the precise mechanisms by which actin microfilaments themselves influence plasmodesmata remain poorly understood. The effect of actin on plasmodesmatal structure and function is complex and varies depending on species, cell type, and experimental conditions [21]. Most functional insights come from assays measuring symplastic transport of fluorescent dyes or proteins, as well as studies involving virus movement proteins, which can themselves manipulate actin to facilitate their passage through plasmodesmata [8].

The prevailing model is that actin at plasmodesmata generally restricts intercellular movement [8]. For instance, mutations in actin-binding proteins or pharmacological depolymerization of actin result in enhanced intercellular transport, while stabilization of actin filaments (e.g., using phalloidin) or mutations in Actin-Depolymerizing Factor 3 reduce transport [8,11,18,22,23]. Nonetheless, such treatments typically affect actin dynamics throughout the cell, not exclusively at plasmodesmata, potentially complicating the interpretation of results [8].

BRK1 is a core subunit of the WAVE/SCAR complex, which is an activator of ARP2/3-mediated branch actin nucleation [24]. BRK1 was first identified in maize, where its loss of function results in unlobed pavement cells resembling bricks, and defective asymmetric cell divisions during stomatal development [25]. In *A. thaliana*, BRK1 carries out similar roles in mediating cell expansion in trichomes and pavement cells, and also plays a role in positioning the region of root hair outgrowth [26]. As with other WAVE/SCAR and Arp2/3 components, loss-of-function mutants frequently have gaps between neighboring cells as a result of weakened intercellular adhesion [27–31].

Here, we investigate the subcellular localization of BRK1-YFP in *A. thaliana* tissues and its relationship to plasmodesmata and primary pit fields, finding that it shows a strong enrichment at cell edges and localizes to plasmodesmata within primary pit fields.

## Results

### BRK1-YFP localizes to cell edges

This work arose from a small exploratory survey to identify proteins that accumulate at cell edges, which are known to house a suite of specialized proteins such as the microtubule-associated proteins CLASP and GCP2/3 [32,33], as well as the endomembrane protein RAB-A5C [34], the receptor-like kinases RLP4/RLP4-L1 [35], and SOSEKI polarity proteins [36]. For this, we obtained and examined several previously published *A. thaliana* lines expressing fluorescent protein-tagged candidate proteins with potential cell edge localization. These included several microtubule-, actin-, and endomembrane-associated proteins. These lines were acquired from stock centers and colleagues, and analyzed by confocal microscopy for evidence of cell edge enrichment. Out of several candidates, BRK1-YFP displayed a strong and distinct enrichment at cell edges (Fig 1), prompting further detailed analysis.

Using plants stably expressing *BRK1::BRK1-YFP* in the *brk1* background [37], we examined several cell types and observed punctate staining around the cell periphery, with notable enrichment at three-way junctions. Three-dimensional imaging revealed that most of the BRK1-YFP signal at the cell periphery corresponds to the cell edges adjoining the outer periclinal faces of two cells (i.e., periclinal cell edges) and the anticlinal edges (which comprise three-way junctions) (Fig 1A-H; S1 Movie). The enrichment BRK1-YFP signal at anticlinal edges showed the strongest accumulation of BRK1-YFP, with the signal typically extended partially or entirely inward, while the periclinal enrichment was more restricted, showing a sharper drop in signal moving inward. Usually, the inner periclinal edges did not show an obvious enrichment compared to the outer periclinal edges (Fig 1).

The pattern of BRK1-YFP signal at cell edges is most clearly exemplified in axially elongating cells, such as those of hypocotyls and roots, where it resembles a horseshoe surrounding the cell edges (Fig 1E, F, H). Interestingly, in these cells the BRK1-YFP cell edge enrichment was often more prominent on transverse walls compared to longitudinal walls.

We also observed BRK1-YFP in leaf mesophyll cells, where it was enriched at the shared anticlinal walls between neighboring mesophyll cells, but unlike epidermal cells, was typically excluded from three-way junctions, both before and after they have separated to form intercellular spaces (Fig 1G).

Our findings are consistent with the observations of Dyachok et al. [37], who described the localization pattern as accumulation along the cell periphery and at cell corners.

### BRK1-YFP localizes to plasmodesmata in primary pit fields

While the bulk of BRK1-YFP localization corresponds to cell edges, we also observed discrete spots along the anticlinal walls in fully expanded cells of the hypocotyl and cotyledons/leaves (Fig 2). Counterstaining with propidium iodide (PI) revealed that many of these puncta were located in circular or elongated regions of low PI signal, which are primary pit fields, where the cell wall is thinner and plasmodesmata are concentrated [5,6]. For comparison, we examined a well-known plasmodesmatal marker, PDLP1-GFP, which was enriched at pit fields but did not show the edge localization characteristic of BRK1-YFP (Fig 3).

To confirm the association of BRK1-YFP with PD/primary pit fields, we stained plants expressing BRK1-YFP with aniline blue, which stains callose at plasmodesmata, late-stage cell plates, and newly formed walls. We observed colocalization between BRK1-YFP and aniline blue, with some exceptions where only one or the other was detectable (Fig 4).

During aniline blue staining, BRK1-YFP signal tended to gradually disappear from its normal locations and became more cytosolic, possibly due to osmotic stress induced by the high glycine concentration in the staining solution. In support of this, plasmolysis with 0.5 M mannitol led to increased cytosolic BRK1-YFP signal (Fig 5; see also [37]). Interestingly, while much of the signal became cytosolic or remained attached to the retracted membrane, some BRK1-YFP punctae remained next to the cell wall, presumably at the base of Hechtian strands (Fig 5).

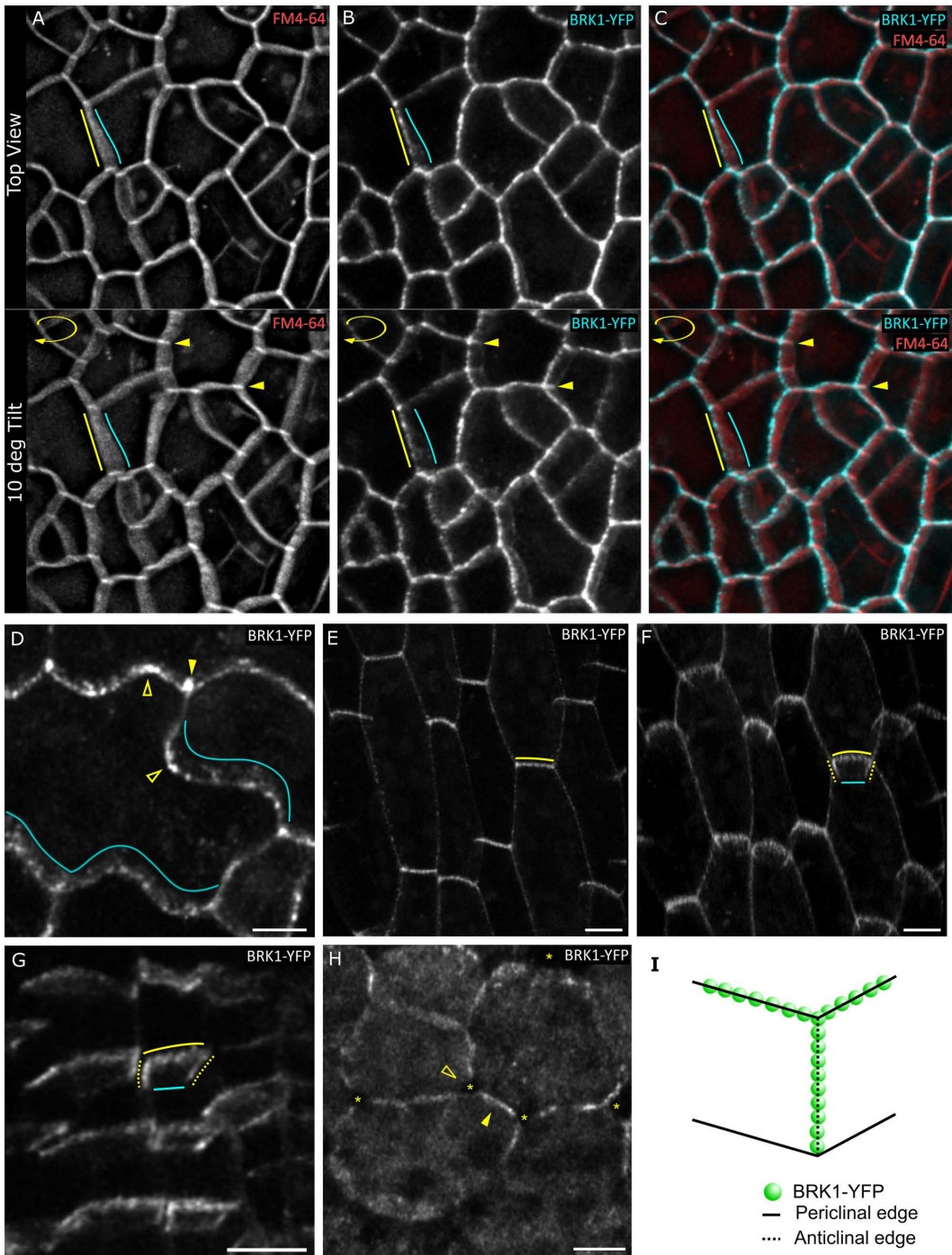

**Fig 1. BRK1-YFP localizes to cell edges in maturing cells. (A-C)** First true leaf, cyan = BRK1-YFP, red = FM4-64, overlapping signal is white. Top panels are projections with no rotation, and bottom panels show the same projection rotated 10 degrees about its y-axis. **(D)** Maturing cotyledon pavement cell showing BRK1-YFP at cell edges (arrowheads) with enrichment at the outer periclinal edges of lobes (hollow arrowheads). **(E-F)** Elongating hypocotyl epidermal cells from top view **(E)**, and tilted view **(F)**. **(G)** Root epidermal cells from early elongation zone. **(H)** Maturing spongy mesophyll cells from a first true leaf. Arrowhead indicates BRK1-YFP signal at shared anticlinal walls between neighbors. Hollow arrowheads indicate absence of BRK1-YFP signal at anticlinal walls bordering intercellular spaces. Asterisks indicate intercellular spaces. **(I)** Schematic illustrating cell edge nomenclature and BRK1-YFP localization pattern. Dotted lines correspond to anticlinal cell edges, and solid lines indicate outer periclinal edges (yellow) and inner periclinal edges (cyan). Scale bar, 5 µm for D and G, and 10 µm for the rest.

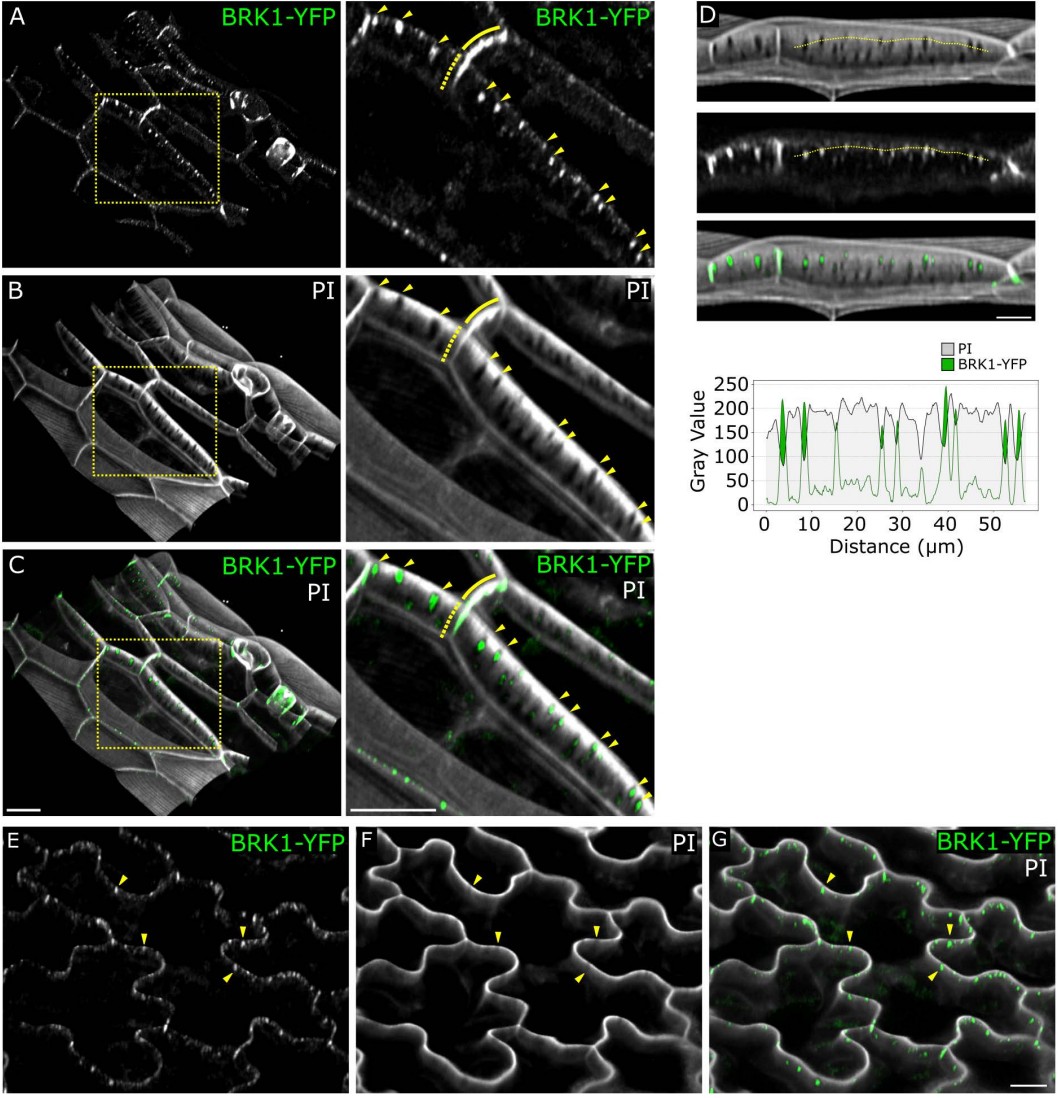

**Fig 2. BRK1-YFP localizes to primary pit fields in mature cells.** (A-C) Mature hypocotyl epidermal cells expressing BRK1-YFP (green) and stained with PI (white). The right panels are enlargements corresponding to the boxed region in the left panels. Solid and dotted lines indicate BRK1-YFP signal at periclinal and anticlinal edges, respectively. (D) Image and corresponding intensity profile plot of anticlinal wall of the outlined cell in A-C, rotated for clarity. The plot corresponds to the dotted line drawn on the image. (E-G) Mature cotyledon pavement cells expressing BRK1-YFP (green) and stained with PI (white). Arrowheads mark BRK1-YFP enrichments at pit fields. Scale bars are 10 μm.

## Discussion

We show here that BRK1 protein localizes prominently to cell edges in several cell types, and additionally accumulates at primary pit fields in mature hypocotyl and leaf/cotyledon cells. This plasmodesmatal localization complements an existing body of evidence implicating actin and actin-binding proteins in plasmodesmata regulation [8], and places WAVE/SCAR complex alongside formins and ARP2/3 as potential key players in plasmodesmatal function.

BRK1-YFP accumulation at plasmodesmata is most evident in mature cells that showed visible pit fields detected as less intensely stained regions of PI along anticlinal walls. While ARP2/3 has been documented at pit fields, it also localizes

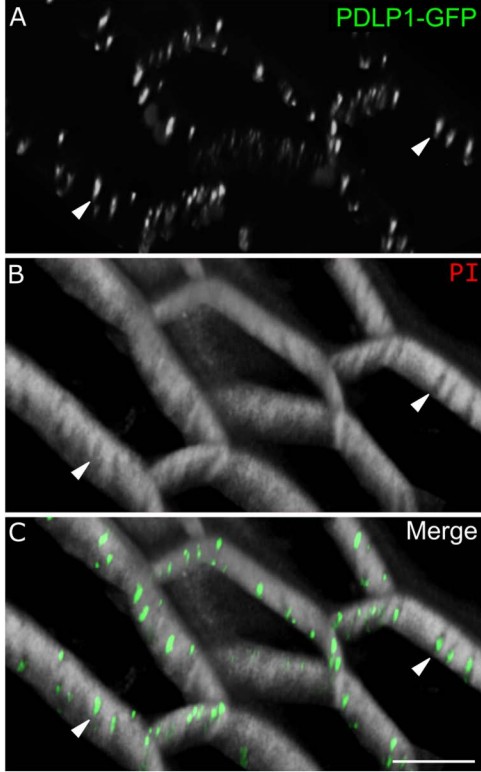

**Fig 3. PDLP1-GFP localizes to primary pit fields but not cell edges. (A)** PDLP1-GFP (green); **(B)** PI (white); **(C)** merged. Arrowheads denote PDLP1-GFP and pit field location. Scale bar is 10 μm.

more broadly to plasmodesmata [13]. Beyond this, most studies identifying actin-binding proteins at plasmodesmata do not differentiate between plasmodesmata and pit fields, leaving gaps in our understanding of how plasmodesmatal composition varies across cell types, developmental stages, and environmental responses.

BRK1 has diverse, species-specific and cell type-specific localization patterns in epidermal cells. In *A. thaliana*, in addition to cell edges and plasmodesmata BRK accumulates at the outer periclinal edge of pavement cell lobes [37], shows enrichment at tips of growing trichomes [37,38], and accumulates at root hair initiation/outgrowth sites [26]. In accordance with these patterns, *A. thaliana brk1* mutants show defective cell lobing, frequent de-adhesion between leaf epidermal cells (often at cell lobes and three-way junctions), distorted trichomes, and aberrant positioning of root hairs [26,28,37]. In maize, BRK1 also shows enrichment at cell lobes and cell edges (referred to as cell corners in [39]), and additionally functions in assembly of the actin patch in subsidiary mother cells, which guides nuclear movement to promote asymmetric cell divisions during stomatal morphogenesis [40–42].

Notably, our observation of BRK1-YFP at cell-cell junctions in mesophyll cells provides evidence for a non-epidermal tissue function for BRK1, which would extend the range of WAVE/SCAR function beyond previous work focusing on epidermal cells. Given the known roles of actin and other actin-binding proteins in modulating intercellular trafficking within the epidermis, it is tempting to speculate that the WAVE/SCAR complex modulates transport within mesophyll tissues as well, but further study is needed.

In summary, our data add a key component of the actin nucleation machinery to the growing body of evidence linking actin and plasmodesmatal functions.

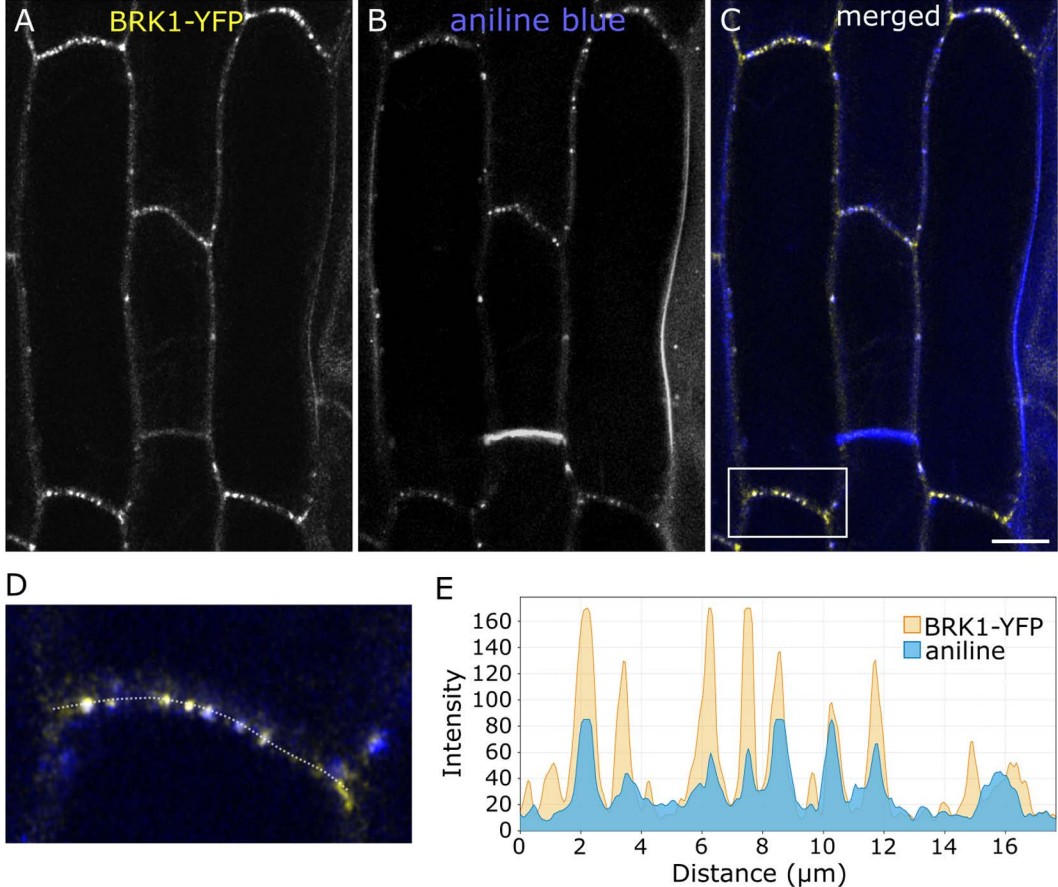

**Fig 4. BRK1-YFP colocalizes with plasmodesmata.** Hypocotyl cells expressing BRK1-YFP **(A)**, stained with aniline blue **(B)**. (C) is merged image of A and B where BRK1-YFP is yellow, and aniline blue is blue. Overlapping signal appears white. **(D)** Enlarged region corresponding to the box in **C. (E)** Fluorescence intensity plot corresponding to the dotted line in **D.** Scale bar is 10 µm.

## Materials and methods

### Plant materials and growth conditions

The transgenic line *BRK1::BRK1-YFP* [37] was kindly provided by Laurie Smith (UC San Diego). Seeds were surface-sterilized with 70% ethanol, rinsed five times with sterile water and plated onto Petri dishes containing ½ MS media, 1.0% agar, 1% sucrose at pH 5.7. Plates were wrapped with parafilm (Bemis Inc.) and placed vertically in a growth cabinet at 22°C, under a 16-h light/8-h dark cycle (cool-white fluorescent tubes; light intensity of ~40 µmol/m$^2$/sec). Roots, hypoco-tyls and cotyledons were imaged at varied times following their germination, ranging from 3 to 7 days after germination, depending on the desired tissue, cell type, and developmental stage.

### Tissue preparation and microscopy

All observations were performed in vivo. For imaging true leaves, cotyledons were excised with fine scissors prior to mounting the specimens. All the tissue samples were mounted in Nunc chambers (Lab-Tek) with 10 µl Perfluoroper-hydrophenanthrene (PP11; Sigma-Aldrich) and covered by 2~3-mm-thick 0.7% Phytagel (Sigma-Aldrich). All confocal images were obtained via a Zeiss Meta 510 with Zeiss Axiovert 200M microscope, 63X water immersion, or Zeiss 880

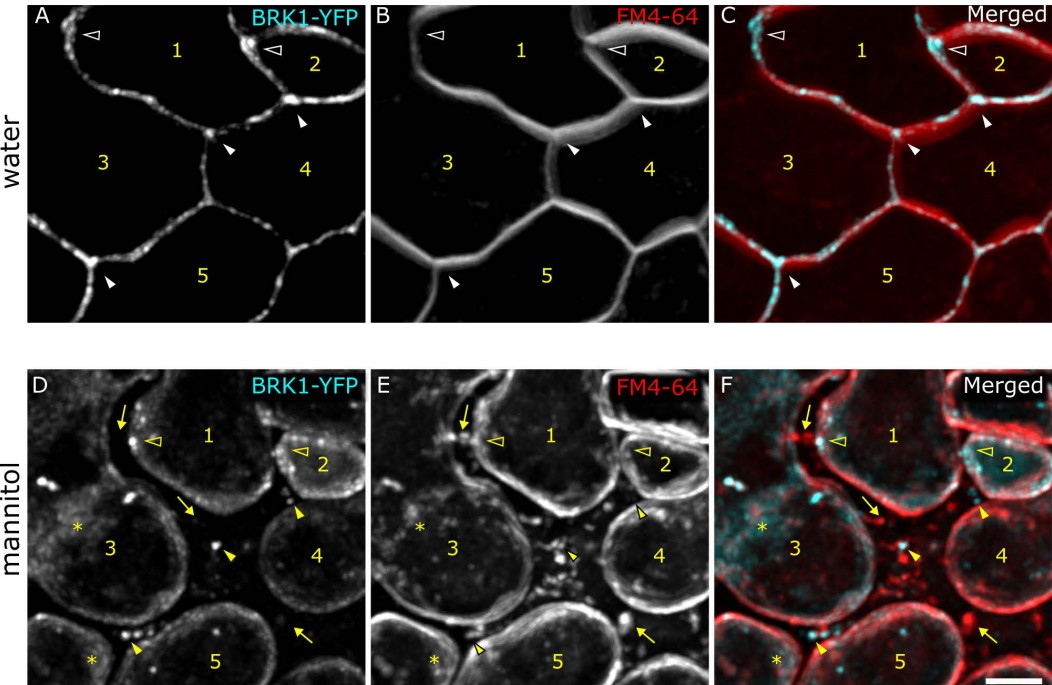

**Fig 5. Cell edge-localized BRK1-YFP associates with retracted plasma membranes, membrane-wall contact sites, and becomes cytosolic after plasmolysis.** Leaf before (A) and after (B) plasmolysis in 0.5 M mannitol for 9 minutes. The same region of the leaf is shown. Cells are numbered for reference, and the positions of arrowheads on each image approximate the corresponding positions before and after plasmolysis. Red is FM4-64-stained membrane and cyan is BRK1-YFP. Solid arrowheads mark the BRK1-YFP dots outside of the retracted plasma membrane next to the cell wall after plasmolysis. Hollow arrowheads indicate BRK1-YFP at the retracted membrane after plasmolysis. Asterisks indicate cytoplasmic BRK1-YFP fluorescence. Arrows mark the FM4-64-stained Hechtian strands. Scale bar is 5 µm.

using the AiryScan detector with both 40X and 63X water immersion objectives. GFP was excited with the 488 nm argon laser line, and fluorescence was collected with a 495–550 nm bandpass emission filter. YFP was imaged using the 514 nm line from an argon laser and captured with emission wavelength of 495–550 nm. FM4–64 and PI were imaged using the 514 nm from an argon laser, with emission captured using a 570–645 nm bandpass filter. For aniline blue, a 405 nm laser was used for excitation, and a 465–505 nm bandpass filter was used for emission. The Z-stack slice intervals varied from 0.2 µm to 0.4 µm. For sampling, more than 5 cotyledons or first pair of true leaves were picked and more than 10 cells for each sample were imaged.

## Image analysis

Images were processed with the ImageJ Fiji distribution [43] (http://rsb.info.nih.gov/ij/). Fluorender was used for 3D renderings [44]. Figures were assembled using Corel Draw software (www.Corel.com; Corel System. Ottawa, ON, Canada), and Inkscape vector graphics software (www.inkscape.org). Data visualization was performed using Python (version 3.8.17) with the Matplotlib (version 3.7.1) and Pandas (1.5.3) libraries.

## PI and FM4–64 staining

To label the plasma membrane, FM4–64 (Sigma-Aldrich) was used. Five-day-old seedlings were incubated with 10 µM FM4–64 in 1.5 ml microfuge tubes and then briefly centrifuged at 3381g for 1 min to allow dye penetration. Stained samples were rinsed with distilled water, mounted with PP11, and imaged in chambers as above.

For cell-wall staining, seedlings were incubated in 100 µg/mL PI (Calbiochem), centrifuged similarly, and incubated for 20 min at room temperature. They were then rinsed, mounted, and imaged as above.

## Aniline blue staining

To visualize plasmodesmata callose, 7-day-old *A. thaliana* seedlings were incubated in aniline blue solution (0.1% aniline blue in distilled water, 1M glycine, pH 8.0) for 20 min at room temperature, rinsed with distilled water and mounted and imaged as above.

## Plasmolysis

For plasmolysis, leaves were excised and mounted in water + FM4–64 between two 24 x 40 mm coverslips separated by strips of vacuum grease on the long edges to create a flow chamber, allowing us to image before and after the addition of mannitol solution (0.5 M), which was wicked from one end as we pipetted it into the other end.

## Supporting information

**S1 Movie. Visualization of BRK1-YFP at cell edges.** Shown is a 3D reconstruction of BRK1-YFP (cyan) and FM4–64 (red) in an *A. thaliana* leaf.
(MP4)

## Acknowledgments

We thank Dr. Laurie Smith for generously providing the BRK1-YFP seeds used for this study.

## Author contributions

**Conceptualization:** Chris Ambrose, Zhihai Chi.

**Data curation:** Chris Ambrose, Zhihai Chi.

**Formal analysis:** Chris Ambrose, Zhihai Chi.

**Funding acquisition:** Chris Ambrose.

**Investigation:** Chris Ambrose, Zhihai Chi.

**Methodology:** Chris Ambrose, Zhihai Chi.

**Project administration:** Chris Ambrose.

**Resources:** Chris Ambrose.

**Software:** Chris Ambrose.

**Supervision:** Chris Ambrose.

**Validation:** Chris Ambrose, Zhihai Chi.

**Visualization:** Chris Ambrose, Zhihai Chi.

**Writing – original draft:** Zhihai Chi.

**Writing – review & editing:** Chris Ambrose, Zhihai Chi.

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
