## [Decision Letter · Decision Letter 0]

27 May 2025

Dear Dr. Ambrose,

Thank you for submitting your manuscript to PLOS ONE. After careful consideration, we feel that it has merit but does not fully meet PLOS ONE’s publication criteria as it currently stands. Therefore, we invite you to submit a revised version of the manuscript that addresses the points raised during the review process.

Thank you for sending your paper to PLoS One. Two reviewers have assessed your paper and while they are both generally supportive they each brought up concerns that should be possible for you to address. Along with requesting more balanced discussion in serval places, reviewer 1 asks for further documentation of the screen that led to looking at brick1. Reviewer 2 makes key points about improving the data presentation and expresses a general ambivalence about the thinness of the paper; that reviewer suggests several ways that could make your paper … thicker. 

I have also read your paper and have some comments of my own. 

The thickness issue is tricky. PLoS One does require a paper to be thick, only valid. You could address the ambivalence of reviewer 2 by adding to the discussion various caveats. For example, you should state that having only one transgenic line is a limitation and likewise no “orthogonal” evidence to back up your findings. But while PLoS One has no requirement for thickness, we—journal and authors—like our papers to have an impact and be cited. Making a paper stronger is intrinsically valuable. One way to do that would be by adding more information about your screen. Doing one or more experiments suggested by reviewer 2 would be another. In my comments here I will also offer a few places where some further work (or perhaps simply further figures from extant work) would be valuable. 

Editor’s comments (note, please respond to these points in the same file as you respond to the reviewers).

Major comments. 

A. Figure 1. You are claiming that figure 1 establishes that brick1 localizes to the outer cell wall edges. But this figure is not convincing in that regard. In Figure 1B, cell wall staining vanishes before the inner edges are reached. How do you know that the brick1 staining also did not vanish for the same reason (i.e., imaging parameters not absence of protein)? In the other cell types, there is no attempt to show inner vs outer cell edges. The paper would be much strengthened by some z-stacks going from outer periclinal wall to inner periclinal wall (and if possible beyond). 

B. Along these lines, you claim in the discussion that brick1 is localized to cell wall junctions in mesophyll and that this localization is a first non-epidermal localization for this protein. However, Figure 1G, where this localization is shown, is not presented in the results specifically. Moreover, without a z-stack, I am not convinced that the imaged cell junctions are mesophyll. How do you know? 

C. In the discussion (Line 149) you write that brick1 is most evident in mature cells. Where is this shown in the results? A developmental time series (or several) showing young, middle aged, and old organs (hypocotyls? cotyledons? leaves?) would make this point and strengthen the paper. 

D. Finally, in figure 1E and F, and also figure 4, brick1 seems to be preferentially localized to transverse walls (compared to longitudinal walls). This looks striking. I am surprised that you don’t comment on this apparent polarity. If it is real and not an accident of the chosen images then that would be worth documenting in a few figures/experiments. 

Minor comments (given in order of occurrence). 

1. Please spell out plasmodesmata. Scientists love acronyms but readers hate them. And, writers ought to hate acronyms too, because they convert a word to a symbol. Doing so impedes thought. Given that plasmodesmata are at the heart of the paper, the writer should think about the language carefully. Write with words not symbols. I am of course aware that PD is standard but so is bad writing. Popularity should not justify avoiding clarity.

1a, Along the lines of promoting clarity I want to add a more speculative suggestion: instead of BRK1, write brick1. The latter is easier to read and looks more handsome on the page. After all, we write actin and not ACTIN. Your protein has a nice name so why not use it? The same can be said for scar/wave. Those are good stout Anglo-Saxon words and look great as such (whereas SCAR/WAVE is an ugly blot). According to arabidopsis lore, genes are supposed to be written in all capital letters; besides that convention differing for other organisms, for the most part here you are writing about proteins. Please consider. 

2. Line 22. Change "Arabidopsis" to either *Arabidopsis thaliana*  (and then *A. thaliana*  after first mention) or arabidopsis. The latter is perfectly correct and would have been the default prior to the advent of molecular biology. A capital letter signifies either a proper name (e.g., Queen Anne's lace) or a genus. Common names are widely taken from the genus (iris, rhododendron) and they are not capitalized or italicized. Unfortunately despite this well established and sensible convention, journals will capitalize Arabidopsis when used as the common name for *A. thaliana* . So the only correct option available to a careful writer is *A. thaliana* .

3. Line 49. Spell out ‘actin-binding proteins’

4. Line 60. You used ‘ADF3’ previously. The convention is to define the acronym at first use. Also while the letters in the acronym are capitalized the letters in what they stand for are not. So it should be written ‘actin-depolymerizing factor 3’. Finally, please consider dropping this acronym altogether. 

5. Line 67. Here you write “Arp2/3” but other places you write “ARP2/3”. You should be consistent. While brick and scar and wave, being words, lend themselves to being spelled out, arp is less ‘word like’. I have no problem with arp2/3 but others might. 

6. Line 74 (and elsewhere). The terminology is confusing. What is a periclinal edge? I think the heading means where the edges between periclinal and anticlinal walls meet. But why not simply ‘outer’ edges? And are you really sure of absence along an edge between two anticlinal walls?

7. Line 76.  Spell out microtubule. And consider writing ‘clasp’ (another good word). 

8. Line 78. No reason to capitalize ‘Soseki’. It is not a person. 

9. Lines 93, 94. I cannot see any solid or dotted lines in Figure 1D.

10. Line 95. What does “maturing” mean? Better to give age and rough position. Dark or light grown? 

11. Line 104 (and elsewhere) spell out propidium iodide. 

12. Line 125 and elsewhere. Do not capitalize aniline blue.

13. Line 132. Change ‘becomes’ to ‘became’ and ‘remains’ to ‘remained’.

We look forward to receiving your revised manuscript.

Kind regards,

Tobias Isaac Baskin

Academic Editor

PLOS ONE

2. Please update your submission to use the PLOS LaTeX template. The template and more information on our requirements for LaTeX submissions can be found at http://journals.plos.org/plosone/s/latex

“NSERC Discovery Grant 2015–05938”

Reviewers' comments:

Reviewer's Responses to Questions

**Comments to the Author**

1. Is the manuscript technically sound, and do the data support the conclusions?

Reviewer #1: Yes

Reviewer #2: Partly

2. Has the statistical analysis been performed appropriately and rigorously?

Reviewer #1: N/A

Reviewer #2: N/A

3. Have the authors made all data underlying the findings in their manuscript fully available?

Reviewer #1: Yes

Reviewer #2: Yes

4. Is the manuscript presented in an intelligible fashion and written in standard English?

Reviewer #1: Yes

Reviewer #2: Yes

Reviewer #1: The manuscript presents a small but solid piece of descriptive, qualitative work that confirms and extends previous studies (most notably Dyachok et al 2008, Ref. 32), providing additional evidence from hitherto uncharacterized cell types, and, more importantly, presents very high quality image documentation surpassing previous reports.

The only substantial weakness I see is lack of clarity with regard to how did the authors pick BRK1 for detailed characterization. They refer to a "screen to identify proteins that accumulate at cell edges" (l. 75) but do not provide any further information how this screen was conducted. Thus, there is a hole in their story which needs to be repaired - either by incuding the description of this screen (which would strenghten the paper substantially!), or by providing a reference to a paper (or preprint) where their screen is reported.

Otherwise there are a couple of less substantial issues that ought to be addressed when revising the manuscript:

1) l. 30-40: The introduction should cite some recent review - there are quite a few (e.g. https://doi.org/10.1111/nph.19666, https://doi.org/10.1093/jxb/erae307).

2) l. 50 - Myosin VIII at plasmodesmata was found much earlier than 2014, see, e.g., https://doi.org/10.1016/s1065-6995(02)00330-x.

3) l. 57-68 - The statement that "in general, the presence of F-actin at PD appears to limit symplastic movement" is an oversimplification, contradicted, e.g., by https://doi.org/10.1007/s00709-010-0244-3. A more balanced discussion of Ref. 18 is needed, such as, e.g., in https://doi.org/10.3389/fpls.2021.647123.

4) In the Discussion, the authors should explicitely acknowledge that similar observations are narratively reported as "data not shown" in Ref. 32.

5) l. 179 - it would be good to include information on light source type

6) l. 200-201 - since no statistical analyses are reported, there is no point describing their methodology!

Reviewer #2: This paper reports a single key finding: BRK1-YFP localizes to plasma membrane puncta. The location of these puncta correlate with a depletion of propidium iodide and deposition of callose, which mark plasmodesmata. Therefore, it is likely that BRK1-YFP localizes to plasmodesmata.

BRK1 is a member of a multi-subunit complex, the SCAR/WAVE complex. While other members of the SCAR/WAVE complex have known functions outside the complex, the only known function of BRK1 is within SCAR/WAVE. The function of SCAR/WAVE is to promote actin nucleation, via ARP2/3 dependent and independent mechanisms.

BRK1 itself has not been directly shown to localize to plasmodesmata, nor (to my knowledge) have other SCAR/WAVE complex members, therefore making the data presented here new. Despite the novelty of this data, it is not surprising, given that members of the ARP2/3 complex have already been shown to localize to plasmodesmata.

I am a bit ambivalent about the data in this paper. The imaging of the single BRK1-YFP transgenic is robust, however only one event has been examined, and there are no orthogonal data to support the observation. Colocalization is shown with aniline blue (but not with PLDP1). I am inclined to believe that BRK1-YFP is enriched at plasmodesmal sites, largely in part because so many other actin-associated proteins are there. If this was the first report of any SCAR/WAVE or ARP2/3 complex member localizing to PD, I would insist on either multiple transgenic events, and/or an independent line of evidence such as a native antibody, or interaction assays with other PD proteins. No functional relevance to this correlation in localization is investigated.

The authors are generally cautious in their language, but none-the-less suggest that (lines 149-150) “BRK1-YFP accumulation at PD is most evident in mature cells, suggesting a specialized function in modulating secondary PD formation or regulating PD permeability (1,33).”

There is no evidence provided here, or elsewhere, that BRK1 or SCAR/WAVE functions to regulate PD formation or permeability. Using the brk1 mutant vs brk1; BRK1-YFP complemented line to assay permeability using a transient transformation assay would suggest a functional role. I believe this is a relatively simple assay. Similarly, quantifying secondary PD in wildtype vs brk1 would address the question of formation, but I think this is a more difficult question, as it would likely require TEM. While normally I am reluctant to ask for additional experiments, I feel like the single result here warrants either some sort of orthogonal support, and/or some of the discussion must be toned down even more.

Regarding data presentation – single channel images should be shown in grey scale to improve visibility in all figures. Green/red combinations are not color blind friendly, and should be adjusted accordingly. It would help readability considerably to (1) use consistent colors for the same construct between figures (e.g, BRK1-YFP is cyan in Figure 1, green in Figure 2, yellow in Figure 4, and green again in Figure 5.). I would also recommend that every panel be labelled with the fluorochrome examined, as in Figure 2. In my initial scan of the paper, I believed Figure 3 (which is PLDP1-GFP ) showed BRK1-YFP. The white lines in Figure 1 that mark periclinal/anticlinal edges are small and hard to notice.

**Do you want your identity to be public for this peer review?** For information about this choice, including consent withdrawal, please see our Privacy Policy

Reviewer #1: **Yes: ** Fatima Cvrčková

Reviewer #2: No

---

## [Author Response · Author response to Decision Letter 1]

8 Jul 2025

Reviewer responses are included as a nicely formatted attachment already.

---

## [Decision Letter · Decision Letter 1]

1 Aug 2025

The Arabidopsis WAVE/SCAR Protein BRICK1 Associates with Cell Edges and Plasmodesmata

PONE-D-25-24278R1

Dear Dr. Ambrose,

We’re pleased to inform you that your manuscript has been judged scientifically suitable for publication and will be formally accepted for publication once it meets all outstanding technical requirements. Thank you for submitting your paper to PLoS One. 

Reviewer one notes two typos for you to fix. I add two other small things:

Line 22 (Abstract) a PD snuck thru! Please change to plasmodesmata.

Line 256 (Acknoledgements) Please add the affiliation of Dr Smith (I think it is University of California at San Diego).

And in bibliographies, words in article titles are not capitalized, even if they were in the journal where the article appeared. That format is reserved for titles of books. Please go thru your reference list and remove such title-word capitalization from the handful of references where it occurs. Of course, words in titles that are proper names, like "Queen Anne's lace", are capitalized for that reason.

Kind regards,

Tobias Isaac Baskin

Academic Editor

PLOS ONE

Additional Editor Comments (optional):

Reviewers' comments:

Reviewer's Responses to Questions

**Comments to the Author**

Reviewer #1: All comments have been addressed

Reviewer #2: All comments have been addressed

2. Is the manuscript technically sound, and do the data support the conclusions?

Reviewer #1: Yes

Reviewer #2: Yes

3. Has the statistical analysis been performed appropriately and rigorously?

Reviewer #1: N/A

Reviewer #2: N/A

4. Have the authors made all data underlying the findings in their manuscript fully available?

Reviewer #1: Yes

Reviewer #2: Yes

5. Is the manuscript presented in an intelligible fashion and written in standard English?

Reviewer #1: Yes

Reviewer #2: Yes

Reviewer #1: This was a nice, rounded piece of work to begin with, and I had only a few minor concerns already in the first version, which were all adequately addressed. (I apologize for the comment on related previous observations in Dyachok et al 2008 not being cited, because I overlooked that they are cited in Results, while I expected them in the Discusssion.) The additional modifications done at the request of the other reviewer and the Editor are further adding value. I am thus recommending acceptance.

I did notice two very minor typos/formal errors that should be corrected at the typesetting stage (no need for requesting a revision):

l. 26 - "plasmodesmata", not "Plasmodesmata"

l, 109-110 should be "enrichment of BRK1-YFP signal"

Reviewer #2: I feel that my comments have been addressed - the phrasing is toned down, as to not directly imply that any functional relevance has been assigned to BRK1. The display items are much clearer now.

A minor point that I have no strong feelings about: an expert in plasmodesma has recently told me that "size exclusion limit" is an old-fashioned/misleading term that shouldn't be used anymore (line 39-40). "Permeability" - which is used in the manuscript - is correct. I am not an expert on plasmodesmata, and honestly have no horse in this race, but I think if you wanted to delete this sentence, it wouldn't harm anything.

I would like to point out to the editor (and authors) that all caps, not italicized, is the standard nomenclature for proteins in A. thaliana. Therefore, BRK1-YFP (and ARP2/3) is correct.

**Do you want your identity to be public for this peer review?** For information about this choice, including consent withdrawal, please see our Privacy Policy

Reviewer #1: **Yes: ** Fatima Cvrčková

Reviewer #2: No

---

## [Editor Report · Acceptance letter]

PONE-D-25-24278R1

PLOS ONE

Dear Dr. Ambrose,

I'm pleased to inform you that your manuscript has been deemed suitable for publication in PLOS ONE. Congratulations! Your manuscript is now being handed over to our production team.

Kind regards,

on behalf of

Dr. Tobias Isaac Baskin

Academic Editor

PLOS ONE